# Use of Antibiotics in Companion Animals from 133 German Practices from 2018 to 2023

**DOI:** 10.3390/antibiotics14010058

**Published:** 2025-01-09

**Authors:** Roswitha Merle, Leonie Feuer, Katharina Frenzer, Jan-Lukas Plenio, Astrid Bethe, Nunzio Sarnino, Antina Lübke-Becker, Wolfgang Bäumer

**Affiliations:** 1Institute of Veterinary Epidemiology and Biostatistics, Veterinary Centre for Resistance Research, Freie Universität Berlin, 14195 Berlin, Germany; katharina.frenzer@fu-berlin.de (K.F.); jan-lukas.plenio@fu-berlin.de (J.-L.P.); nunzio.sarnino@fu-berlin.de (N.S.); 2Institute of Pharmacology and Toxicology, Freie Universität Berlin, 14195 Berlin, Germany; leonie.feuer@fu-berlin.de (L.F.); wolfgang.baeumer@fu-berlin.de (W.B.); 3Institute of Microbiology and Epizootics, Veterinary Centre for Resistance Research, Freie Universität Berlin, 14163 Berlin, Germany; astrid.bethe@fu-berlin.de (A.B.); antina.luebke-becker@fu-berlin.de (A.L.-B.); 4German Environment Agency, Wörlitzer Platz 1, 06844 Dessau-Roßlau, Germany

**Keywords:** antimicrobials, dogs, cats, consumption, critically important antimicrobials, AMU

## Abstract

**Background/Objectives**: While antibiotic usage in farm animals has been systematically monitored and reduced in many countries, including Germany, data on companion animals such as dogs and cats remain scarce. To address this gap, a study was conducted in Germany to analyze patterns of antibiotic use in dogs and cats. **Methods**: Antibiotic usage data were obtained from debevet, a cloud-based veterinary practice management software based in Berlin, Germany. Practices with fewer than 100 patients were excluded, and data from 2018 to 2022 were analyzed. **Results**: The analysis included 477,310 consultations of 78,381 dogs and 241,532 consultations of 55,729 cats across 133 veterinary practices. Antibiotics were used in 12.9% of dog consultations and 22.5% of cat consultations, with substantial variation across practices. Aminopenicillins, particularly amoxicillin, were the most commonly used antibiotics, while the highest-priority critically important antibiotics (HPCIAs) accounted for 12.4% of treatments. Follow-up treatments led to changes in antibiotic substances in 9.3% of cases, often within the first two days. Indications varied by species, with respiratory issues more frequent in cats and orthopedic problems in dogs. Body weight and breed characteristics influenced the likelihood of antibiotic treatment, with short-nosed breeds showing higher odds. **Conclusions**: The routine data analysis provided valuable insights into antibiotic use in dogs and cats, enabling tracking trends across species and indications over time. While some specific information was missing, the invoicing data’s completeness, the cost-effectiveness of their use, and their unbiased nature make them a robust tool for monitoring and informing legislative changes.

## 1. Introduction

The use of antibiotics is the main driver for the selection of resistant bacteria [1,2]. In veterinary medicine, for the most part, the antimicrobial substances used are the same as those used in human medicine [3]. Thus, the use of antibiotics in animals may contribute to the occurrence of infections with resistant bacteria in humans, which are difficult to treat [4,5]. The World Health Organization (WHO) has published a list of substances that are critically important for human health and, thus, should be restricted to human medicine [6].

In farm animals, the use of antibiotics is recorded systematically in many countries, and in Germany, a surveillance system in farm animals came into force in 2014 [7]. Since their introduction, their usage has decreased from 1238 tons in 2014 to 529 tons in 2023 [8].

Little is known about the use of antibiotics in companion animals such as dogs and cats. In a study from the UK between 2012 and 2014 by Buckland et al. [9], it was reported that 25% of dogs and 21% of cats seen at veterinary practices received antibiotic treatment at least once. Another publication including studies from Belgium, Italy, and The Netherlands reported that 19% of cats and dogs received at least one antimicrobial treatment in the preceding 6 months in 2020 [10]. The average treatment length was 1.8–3.3 days. Similar values were reported from sentinel veterinary practices in the UK, with 18.8% of dogs and 17.5% of cats receiving at least one course of antibiotic treatment [11]. A study in Italy revealed that 38% of all cats presented to the Veterinary Teaching Hospital in Pisa were treated with antimicrobials and that there was a significant decrease between 2017 and 2022 [12]. In a German teaching hospital, more than 50 percent of dogs and cats were given antimicrobials [13] and a study from the US reported 36.5 percent of treated dogs and cats in 14 veterinary teaching hospitals in 2020 were given antimicrobials [14].

To date, no continuous monitoring of usage data in companion animals exists, although this would be crucial to observe any reduction. The findings of Hopman et al. [15] perhaps show that the implementation of antimicrobial stewardship programs could lead to a 15% reduction in antibiotic use in terms of Defined Daily Doses within two years.

To prepare for the possible implementation of a monitoring system in Germany, a feasibility study was conducted on behalf of the German Ministry of Food and Agriculture (support code 2820HS002). The data collected in the context of this study were analyzed with the following aims: (i) overview of the use of antibiotics in cats and dogs overall and per practice and, (ii) usage patterns per substance including highest-priority critically important ones, (iii) per indication, (iv) per age, and (v) per breed.

## 2. Results

In total, 477,310 consultations of 78,381 dogs and 241,532 consultations of 55,729 cats from 133 veterinary practices were included in the analysis; 130 practices treated dogs (16–4285 patients, median 223), and 95 treated cats (11–2847 cats, median 354). One or more antibiotics might have been used or delivered during one consultation, e.g., one injection during the consultation and delivery of tablets for further treatments at home. In total, antibiotic treatment took place in 115,911 consultations, 61,677 of which being for dogs and 54,234 being for cats. In 29,934 consultations, two or more antibiotics were applied or delivered.

In 12.9% of dogs’ and 22.5% of cats’ consultations, antibiotics were applied or delivered. The values per practice were between 0 and 27% (dogs), and 82% (cats), and averaged 9.0% and 18.7%, respectively (Figure 1). Eleven practices never treated cats with antibiotics, and twenty-five practices never treated dogs within the investigated time period.

In total, 74.3% and 66.1% (dogs and cats, respectively) of all treatments were defined as initial treatment. In 23.1% and 38.7% of these initial treatments, a second consultation with antibiotic treatment took place within seven days; 15.4% of all dogs and 18.6% of all cats received at least antibiotic treatments two times within one year.

The percentage of animals that returned to the practice in the first four days was higher for animals with antibiotic treatment than for animals without antibiotic treatment. After day 5, animals without treatment returned more frequently (Figure 2).

Of the 115,911 consultations with treatment, a human pharmaceutical was used 5117 times, i.e., at a rate of 4.4%. The majority of these drugs were administered locally (95.0%), e.g., as eye or ear drops. Gentamicin was used in 63% of these cases, followed by ofloxacin (13%) and oxytetracycline (8%).

### 2.1. Antibiotics Use per Substance Class

Aminopenicillins comprising amoxicillin, ampicillin, and benzylpenicillin were the most commonly used substance class in cats and dogs. Amoxicillin without clavulanic acid was used in 27% and 49% of all treatments in dogs and cats, respectively, and amoxicillin with clavulanic acid was used in 32% and 23% of all treatments. Highest-priority critically important antibiotics (HPCIAs, [6]) were used in 12.5% of treatments, with similar distributions in cats and dogs (Figure 3).

Some consultations comprised treatments with more than one substance. This was the case in 13,373 consultations, i.e., 11.5%. In 695 consultations, three substances, and in 42 consultations, four substances were used. In 7293 consultations (55% of treatments with more than one substance), amoxicillin was combined with amoxicillin and clavulanic acid. Amoxicillin and enrofloxacin were combined in 709 consultations (492, only amoxicillin; 217, amoxicillin and clavulanic acid) and often associated with the urogenital tract (13%). The combination of amoxicillin with gentamicin was used in 591 consultations (396, only amoxicillin; 195, amoxicillin and clavulanic acid), which were mostly related to ophthalmologic (30% of consultations) or otologic disorders (15%). Amoxicillin and neomycin were combined in 391 consultations (191, only amoxicillin; 200, amoxicillin and clavulanic acid), which were often related to dermatology (25%) or the urogenital tract (12%), and 364 consultations (193, only amoxicillin; 171, amoxicillin and clavulanic acid) using amoxicillin and polymyxin were mainly associated with dermatology (29%) and otology (20%).

In 2664 cases (13.0%, dogs: 18.1%, cats: 9.3%), follow-up visitations led to a change in the antibiotic substance. In 30% of cases, the change took place on the first day after initial treatment, and in 48% of cases, it took place within the first two days. Most frequently, amoxicillin and clavulanic acid were used following an initial treatment with penicillin (118 cases). Enrofloxacin was used after initial treatment with amoxicillin (147 cases). The change from amoxicillin or amoxicillin with clavulanic acid to any fluoroquinolone took place 256 times, out of which the first 139 instances occurred the first two days after the initial treatment.

### 2.2. Antibiotics Use per Indication

Of all consultations, 311,459 (43.3%) could be assigned to an indication group. Figure 4 shows the distribution of the various indications given during veterinary consultations. Diagnostics/therapy, dermatology, orthopedics, and digestive system were the most frequently assigned categories. Differences between dogs and cats were mainly seen in orthopedics and in otologic problems, which occurred more frequently in dogs than in cats. The respiratory system was more often affected in cats than in dogs.

Short-nosed dog breeds had 58% higher odds of respiratory disorders (Table 1). Breeds with a predisposition to skin problems had slightly increased odds, at 18% higher, for consultations due to dermatologic disorders than other breeds. The chance of ophthalmologic diagnoses was 24% higher in breeds that often develop entropy or ectropion.

The percentage of consultations with antibiotic treatment varied between the indication groups, with a total of 29.2% in cats and 17.0% in dogs. The differences between cats and dogs were predominant in cases of emergency, orthopedics, trauma, intoxication, and the respiratory system (Figure 5).

### 2.3. Antibiotics Use per Age

The highest percentage of consultations with antibiotic treatment was recorded for animals up to ten years of age (Figure 6).

### 2.4. Influence Factors for the Use of Antibiotics in Dogs

The probability of being treated with antibiotics is dependent on the body weight (*p* < 0.0001). Animals with a higher body weight had a higher chance of treatment. This effect becomes obvious from a body weight of approx. 50 kg (Figure 7). Crossbreeds had a slightly lower probability for antibiotic treatment (12.4%, n = 115,345 consultations) than pure breeds (13.1%, n = 361,965 consultations). Breeds with short noses had 20% higher odds for antibiotic treatment than other breeds in general, but concerning the treatment of respiratory disorders, the odds for treatment were not higher (Table 2 and Table 3).

Breeds with a predisposition to skin problems had only slightly elevated odds of receiving antibiotic treatment. Breeds that typically develop entropy or ectropion problems had a 17% higher chance of receiving antibiotic treatment, although they were more often not treated with antibiotics due to ophthalmologic problems than other breeds (Table 2 and Table 3).

## 3. Discussion

Dogs and cats in Germany are treated with antibiotics on a regular basis, with cats having significantly more treatments than dogs. The percentage of consultations with treatment varies between indications and is high in cases of emergency, trauma, and ophthalmologic as well as respiratory diseases. Aminopenicillins were used in two-thirds of all treatments.

As far as we know, this is the largest study on antibiotic use in relation to the number of veterinary practices and consultations that have been published in Germany. Singleton et al. [11] included more than 600,000 dogs and cats in their study in the United Kingdom and found higher percentages of antibiotic treatment for dogs, but not for cats. Joosten et al. [10] also reported higher values but included only 303 animals from different countries. Three other studies also reported higher values because they only included teaching (secondary) hospitals that use more antibiotics than outpatient practices [12,13,14]. It remains unclear why the practices in our study used fewer antibiotics than in the other studies.

It remains also unclear why the variation in antibiotic treatment between the practices was so great. We could neither differ between hospitals and practices nor between first-, second-, and third-line practices, which is a weakness of our study and would be useful information for future studies. Hopman et al. [15] showed that the usage patterns differed between the types of practices and hospitals.

The percentage of practices that never treated dogs or cats was surprisingly high, although we thoroughly excluded non-veterinary practices such as naturopaths and physiotherapists, and although they had at least 105 and up to 1294 consultations.

The difference in the percentage of consultations with antibiotic treatment between cats and dogs might be due to the fact that cats show fewer clinical signs than dogs; thus, cat owners only become aware of feline disease at a more severe stage of illness and more severely ill cats are presented to veterinarians than dogs. It was striking that follow-up treatments were more frequent in cats than in dogs, although the antimicrobial substance was changed more often in dogs than in cats. The follow-up treatments in cats might also be linked to the worse health status of cats presented to the veterinarian. It is also reasonable why follow-up consultations tended to take place earlier in animals treated with antibiotics: While non-infectious diseases often require mid- to long-term treatment before the effectiveness of the therapy becomes evident, the impact of antibiotic treatments typically becomes clear within a few days. If the antibiotic is ineffective, owners are likely to return to the veterinarian sooner for a follow-up consultation. Animals whose condition deteriorated were re-presented as early as the next day, which also occurred more frequently in the treated animals than in the non-treated ones.

The percentage of treatments with human pharmaceuticals was in the same range as that reported by Singleton et al. [11] who found a value of 8.2% of dogs and 1.7% of cats being administered antibiotics registered for human use. The main reason to use human pharmaceuticals is that no products for veterinary use are on the market.

The predominant use of aminopenicillins (mainly amoxicillin with or without clavulanic acid) in dogs and cats was higher than that in any other study [11,13,14,16]. The German Regulation of Veterinary Pharmacies, which was revised in 2018, aims to promote the prudent use of antimicrobials [17]. Antimicrobial susceptibility testing is mandatory for the use of third-/fourth-generation cephalosporins and fluoroquinolones. In consequence, many veterinarians in Germany always use amoxicillin as a first-line treatment and eventually switch to another substance [18]. The moderate percentage of treatments with HPCIAs might also be the result of this amendment, although we did not see a substantial decline in antibiotic use over the years in our study (Appendix A). Joosten et al. [10] reported 33% of treatments with HPCIAs in 2015, while Singleton et al. [11] found more than 35% of treatments involved usage of third-generation cephalosporins in cats from 2014 to 2016. The analysis of insurance data by Hardefeldt et al. [19] revealed that only 8% of treatments included HPCIAs and that cats were also very frequently treated with third-generation cephalosporins. In a teaching hospital in Hannover, Germany, only 8.2% (2017) and 9.5% (2018) of treatments were HPCIAs with quinolones being the substance class of highest use [13]. In this study, a lot of nitroimidazole was used (23% of treatments in dogs, 19% in cats), while this substance class was used only rarely in our study.

The use of several substances on the same date was not reported in any other study, and in our view, the percentage of 5.5% was surprisingly high. In most cases, local treatment was combined with either parenteral or oral treatment with different substances. Thus, this might have been due to the availability of certain substances for local administration.

The change from aminopenicillins to fluoroquinolones can be regarded as a typical treatment respecting the German legislation when initial treatment with amoxicillin failed and was replaced by enrofloxacin after antimicrobial susceptibility testing. Although our data only allow speculations about the situation, this might indicate that there is a need to train veterinarians concerning the proper procedure for changing the substance.

Unfortunately, we could assign an estimated indication only in 43% of all consultations, and even these assignments were questionable in some of our random manual inspections. An improvement in the methods of automatic categorization is necessary and will be discussed elsewhere. Assuming that the correctness of the allocation is independent of the disorder itself or the antibiotic treatment, the results are discussed in the following paragraphs.

The distribution of consultations over the categories was similar to what was reported by Hardefeldt et al. [19] and Singleton et al. [11]. High percentages of consultations with antibiotic treatment were detected in indications that are typically associated with bacterial infections, e.g., ophthalmology and otology, respiratory and urogenital diseases, or the digestive system, but also in the case of emergency and trauma. The high use of antibiotics in cats in cases of emergency or trauma and due to respiratory disorders indicates that the health status of cats is much worse when they are presented to the veterinarian compared to dogs.

Although predispositions of certain dog breeds are widely described, we could not find any investigation concerning the higher use of antibiotics in these breeds except rather specific reports (e.g., [20]) or general recommendations (e.g., [21]). We were able to clearly show that the respective breeds had higher odds for consultations due to the specific indications and because they received treatments more often than other breeds did, although there was no such relationship for treatment regarding the respective indication. This could lead to the conclusion that these breeds could be more susceptible to infections overall. The comparison between crossbreeds and pure breeds has also shown that the former ones received antibiotic treatment less often, which fits the assumptions that crossbreeds have better general fitness [22].

Neither the role of animal age nor the impact of dogs’ body weight has been addressed in the context of the use of antibiotics. While the increase in antibiotic therapy for animals with (too) low or (too) high weights seems reasonable, it is important to note that the weights used in our analysis are based on the average weights of the breeds, not the actual weights of the animals. Larger animals naturally require higher doses of antimicrobial agents due to their body mass, which could explain part of the observed trend. This limitation allows us only to observe that particularly larger dog breeds tend to receive more antibiotics than small and medium-sized breeds. The decrease in antibiotic treatment for older animals, however, did not correspond to our expectations and remains unexplained. Possible reasons might be that older dogs are more likely to develop neoplasia and other chronic diseases that are not treated with antibiotics, so the absolute number of treatments with antibiotics is the same, but the relative percentage decreases.

One limitation of the study is that we used routine data with missing data in some entries (e.g., no anamnesis or diagnosis recorded) and with imprecise recordings (e.g., “amoxicillin” was given as the drug name, but no implication of the trade name or the concentration). Some specific information concerning diagnoses, the correct trade name of the pharmaceutical, or the duration of the treatment would have enabled a more thorough assessment of the reasons for the use of antibiotics. On the other hand, it would hardly have been feasible to collect so much data explicitly for a study.

Another drawback of the study is that the validity of the study results concerning all German practices could not be investigated, because information on geographical distribution, the number of patients per practice and the number of first-, second-, and third-line practices in Germany is not known. Since the study included small to large practices all over Germany, we are not aware of any reasons that would preclude representativeness.

In addition, the allocation of the free-text entries into an indication category was only successful in less than half of the consultations. In the future and with the use of artificial intelligence, the quality of text mining methods will increase and will lead to better and more precise results.

## 4. Materials and Methods

This study was approved by the Ethics Commission of Freie Universität Berlin, Germany, under the number ZEA-Nr. 2021-018.

Data on the use of antibiotics were provided by debevet, a cloud-based practice management software located in Berlin, Germany. Data included anonymized practice_id, as well as basic information about the patient (species, breed, sex, castration status, date of birth, date of death) and the date of the consultation. Anamnesis and diagnoses were recorded as free text. Each row represents an entry on the invoice, which can either be a treatment with a medicine or consumables, or an examination that is billed for. All entries from the same date and the same animals were regarded as one consultation that could either include antibiotic treatment or not. Any application or delivery to the owner of antibiotics was regarded as treatment. Throughout the text, the term treatment refers to antibiotic treatment.

Free texts in the fields of anamnesis and diagnoses were compared with a list of >25,000 keywords of diagnoses, treatments, and localizations and categorized into one out of twenty indication categories using Sklearn CountVectorizer (version 1.6). The category “diagnostics/therapy” included keywords concerning vaccination, castration, euthanasia, examination, treatment, prophylaxis, etc. “Endocrinology” comprised diabetes, thyroid, Cushing, and others. “Routine” covered mainly parasitic treatments, but also “delivery of drugs” and chipping. “Systemic” included weakness, anorexia, fever, and other generic disorders, while terms that did not allow any specification such as “ultrasound” or “edema” were categorized as “unclear”.

Animal weights were not given in the data, but the average animal weight of the respective breed was used for dogs. If the breed was missing, 20 kg was used as the animal weight (17.1% of consultations with dogs). For cats, an average weight of 5 kg was assumed.

Information about the pharmaceutical product that was applied or delivered was added to each consultation with antibiotic treatment and included the amount per package, the name and concentration of the substance(s), and the application form. For veterinary pharmaceuticals, information was retrieved from vetidata.de (accessed on 15 April 2024), and for human drugs, it was obtained from the “Rote Liste” [23].

Practices with less than 100 patients were excluded from the data analysis. Only the years 2018 to 2022 were included.

HPCIAs were defined following WHO’s list of medical substances [6]. They comprise third and fourth-generation cephalosporins, fluoroquinolones, macrolides, and polymyxins.

### Statistical Analyses

Data were analyzed using R version 4.4.1 with RStudio version 2024.04.2, including the packages “tidyverse”, “here”, “ggplot2”, “scales”, “emmeans”, and “mgcv” [24,25,26,27,28,29,30,31].

The percentage of consultations with antibiotic treatment was calculated for cats and dogs separately. Cases without treatment during the 30 preceding days were regarded as initial treatment, while treatments that took place within 7 days after another treatment were defined as follow-up treatments. The analysis of follow-up investigations with a change in substance was restricted to consultations with only one substance per consultation, resulting in 20,519 consultations with initial treatment.

Non-linear logistic regression models with natural splines and 4 degrees of freedom were applied to investigate the influence of body weight and the age of dogs on the probability of antibiotic treatment. Also, the effect of certain breeds on the occurrence of specific indications and the influence of these breeds on the probability of being treated with antibiotics in general as well as concerning specific indications were analyzed in simple logistic regression models. *p*-values, OR, and the respective 95% CIs are reported.

For the analysis of the effect of crossbreed dogs, all entries that contained the term “Mischling” as a breed, were defined as crossbreeds, and all others were defined as pure breeds. The breeds Affenpinscher, Boston Terrier, Bordeaux dog, Boxer, Cavalier King Charles Spaniel, Chihuahua, English Bulldog, French Bulldog, Griffon, Lhasa Apso, pug, Norwich Terrier, Pekinese, Shi-Tzu, Toy Spaniel, Yorkshire Terrier, and Pomeranian were summarized as short-nosed breeds [32,33]. In addition, American Staffordshire terrier, dachshund, Doberman pinscher, great dane, French bulldog, Labrador retriever, greyhound, Irish setter, poodle, Yorkshire terrier, basset hound, bloodhound, bulldogs, bullmastiff, Neapolitan mastiff, pug, shar pei, and Pekinese were categorized as being predisposed to skin problems. Finally, basset hound, Bernese mountain dog, St. Bernard, bloodhound, bulldog, Cocker Spaniel, Newfoundland dog, Shar Pei, bull terrier, Chow-Chow, poodle, and Rottweiler were grouped, because of their increased risk for entropy or ectropium.

## 5. Conclusions

In conclusion, the use of routine data from 133 veterinary practices over a six-year period provided unprecedented insights into the use of antibiotics in dogs and cats in Germany. The data allowed us to separate between substances and allocate treatments to indication categories. This allows us to track changes between species or over time and thus, to follow up changes in legislative regulations.

Although there were some disadvantages to using routine data, as we lacked some specific information, the fact that there was a large amount of data that were available at a low cost outweighed the disadvantages. And since veterinarians documented the treatments in order to create an invoice, the data were also complete and unbiased. With active monitoring, veterinarians might be more tempted to manipulate the data.

## Figures and Tables

**Figure 1 antibiotics-14-00058-f001:**
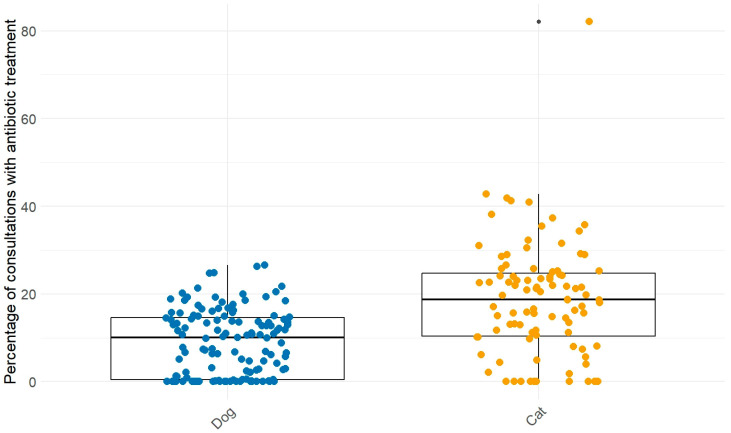
Percentage of consultations with antibiotic treatment per veterinary practice. The ● represents an outlier outside the range of 1.5 standard deviations.

**Figure 2 antibiotics-14-00058-f002:**
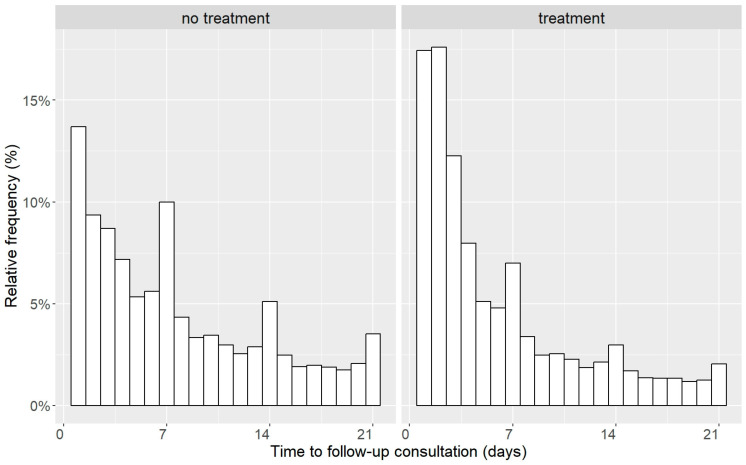
Histogram of time to follow-up consultations for dogs and cats, categorized by consultations with or without treatment with antibiotics.

**Figure 3 antibiotics-14-00058-f003:**
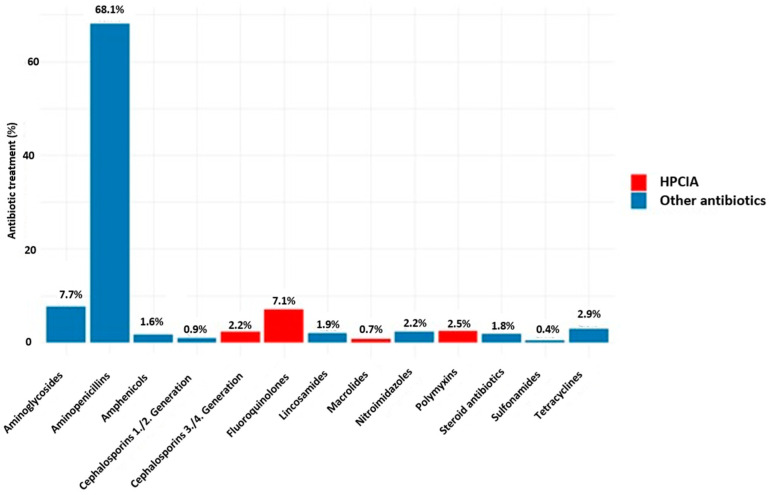
Percentage of consultations with antibiotic treatment per substance class in dogs and cats (145,846 applications or deliveries). Highlighted are substance groups that belong to the highest priority critically important antibiotics.

**Figure 4 antibiotics-14-00058-f004:**
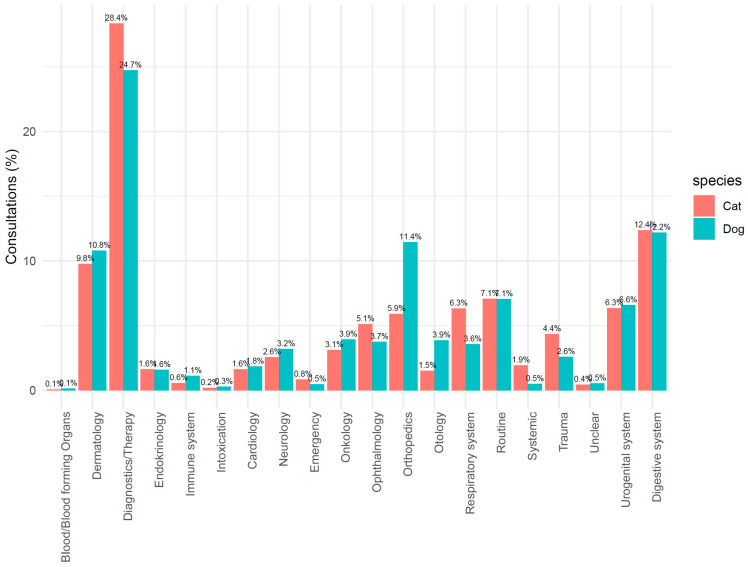
Percentage of consultations per indication group in cats and dogs.

**Figure 5 antibiotics-14-00058-f005:**
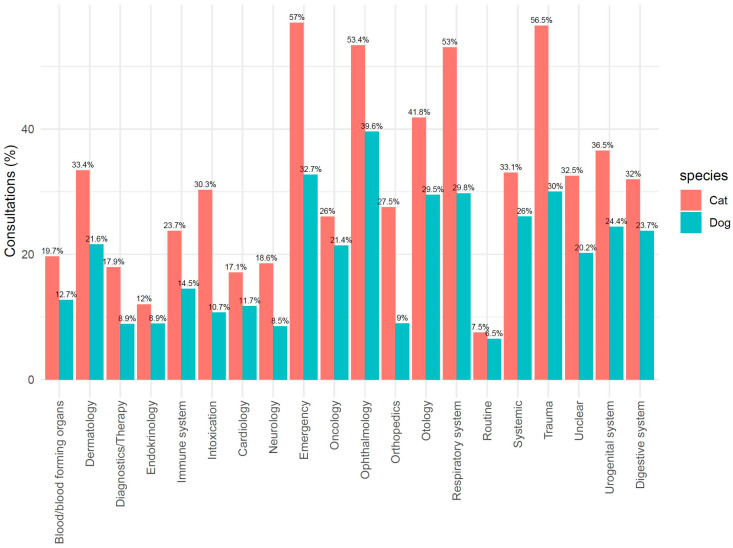
Percentage of consultations with antibiotic treatment per indication group in cats and dogs.

**Figure 6 antibiotics-14-00058-f006:**
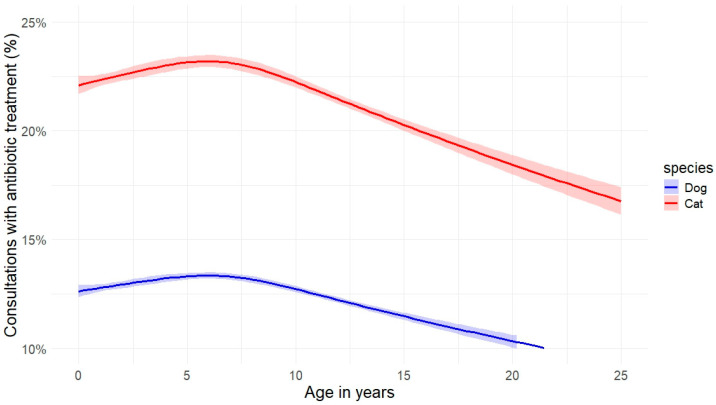
Percentage of visits with antibiotic treatment in cats and dogs over age. Displayed are the estimates of a non-linear regression model with natural splines, 4 degrees of freedom, and a 95% confidence interval; n = 610,854 consultations.

**Figure 7 antibiotics-14-00058-f007:**
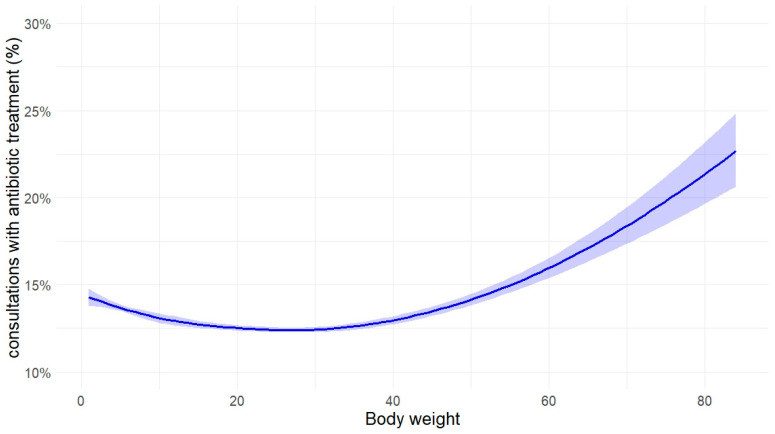
Influence of body weight on the probability of being treated with antibiotics in dogs (n = 477,310 consultations). Displayed are the estimates of a non-linear regression model with natural splines, 4 degrees of freedom, and a 95% confidence interval.

**Table 1 antibiotics-14-00058-t001:** Occurrence of certain indications in canine breeds with specific predispositions to health problems. Results of logistic regression models with the predisposition as an independent variable and the indication as a dependent variable. OR = odds ratio.

Predisposition	Indication Group	Predisposition Present	Indication Frequencies n (%)		OR	95% Confidence Interval	*p*-Value
			No	Yes			
short nose	respiratory system	no	425,964	6400 (1.5)	1.58	1.48–1.69	<0.0001
		yes	43,904	1042 (2.3)			
skin	dermatology	no	377,955	18,203 (4.6)	1.18	1.14–1.22	<0.0001
		yes	76,797	4355 (5.4)			
Entropy or ectropion	ophthalmology	no	448,368	7371 (1.6)	1.24	1.13–1.37	<0.0001
		yes	21,139	432 (2.0)			

**Table 2 antibiotics-14-00058-t002:** Antibiotic treatment of canine breeds with specific predispositions to health problems. Results of logistic regression models with the predisposition as an independent variable and antibiotic treatment as the dependent variable. OR = odds ratio.

Health Problem	Health Problem Present	Treatment Frequencies n (%)		OR	95% Confidence Interval	*p*-Value
		No	Yes			
short nose	no	377,326	55,038 (12.7)	1.20	1.17–1.23	<0.0001
	yes	38,250	6696 (14.9)			
skin	no	345,312	50,846 (12.8)	1.05	1.03–1.08	<0.0001
	yes	70,264	10,888 (13.4)			
entropy, ectropium	no	397,185	58,554 (12.8)	1.17	1.13–1.22	<0.0001
	yes	18,391	3180 (14.7)			

**Table 3 antibiotics-14-00058-t003:** Antibiotic treatment of canine breeds with specific predispositions to health problems within the related indication. Results of logistic regression models with the predisposition as an independent variable and antibiotic treatment as a dependent variable. OR = odds ratio.

Health Problem	Indication	Health Problem Present	Treatment Frequencies n (%)		OR	95% Confidence Interval	*p*-Value
			No	Yes			
short nose	respiratory system	no	14,234	3969 (21.8)	0.94	0.86–1.02	0.123
		yes	3452	903 (20.8)			
skin	dermatology	no	4812	1779 (27.0)	0.98	0.86–1.11	0.722
		yes	1046	435 (29.4)			
entropy, ectropium	ophthalmology	no	4451	2920 (39.6)	0.98	0.80–1.19	0.838
		yes	263	169 (39.1)			

## Data Availability

The data presented in this study are available on request from the corresponding author. The data are not publicly available due to privacy restrictions.

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
