# Peer review of "Use of Antibiotics in Companion Animals from 133 German Practices from 2018 to 2023"

_antibiotics, 2025, doi:10.3390/antibiotics14010058_

Round 1

Reviewer 1 Report

Comments and Suggestions for Authors

The research manuscript reports the data interpretation on use of antibiotics in companion animals especially dogs and cats in Germany. Manuscript research is interesting and limited work has been done in this regard. Research is well presented nd have valuable information regarding future perspective in treatment of companion animals. However, needs some improvements as suggested below. 

1. In first line of introduction authors stated that “The use of antibiotics is the main driver for the selection of resistant bacteria”. Here authors are advised to discuss more about antibiotic resistance emergence and issues. Following studies are suggested to be added here 

Kirn N, Lu S, Aqib AI, Akram K, Majeed H, Muneer A, Murtaza M, Prince K, Li K, 2023. Annotating susceptibility potential of single, double, tri and tetra mixed infection bacteria against non-beta lactam antibiotics. Pak Vet J, 43(3): 596-600. http://dx.doi.org/10.29261/pakvetj/2023.060   

Riaz A, Shahzad MA, Ahsan A, Aslam R, Usman M, Rasheed B, Hussain M, Irtash M, Saleemi MK, Ali A, Sajid SM and Irshad H, 2023. Investigations into the role of zoo animals in transmitting the extended spectrum beta lactamases (esbl) E. coli in the environment. International Journal of Veterinary Science 12(6): 832-837. https://doi.org/10.47278/journal.ijvs/2023.046. 

Li X, Zhu X, Xue Y, 2023. Drug resistance and genetic relatedness of Escherichia coli from mink in Northeast China. Pak Vet J, 43(4): 824-827. http://dx.doi.org/10.29261/pakvetj/2023.062.

2. How the authors ensure the Validity of data collected which were parameters to authenticate the collected data and sources on antibiotics use are free from errrors?

3. In discussion section where authors have extensively discussed the data interpretation of various classes of antibiotics use in dogs and cats. At line 271 authors stated that the high use of antibiotics in cats in cases of emergency or trauma and due to respiratory disorders indicates that the health status of cats is much worse when they are presented to the veterinarian compared to dogs. here authors are advised to discuss little about  the adverse effects of synetic drugs and outcomes of tretamnet in companion nanimals following studies be added here. 

Bielik R, Korim P, Kašelák I, Lukáč B and Bielik B, 2023. Thrombocytopenia in a dog due to long-term administration of phenylpropanolamine: A case report. International Journal of Veterinary Science 12(6): 897-899. https://doi.org/10.47278/journal.ijvs/2023.028

Iqbal N, Khan MA, Luqman Z, Rashid HB, Khan MUR, Hussain N, Aslam S and Ali HM, 2021. A biochemical study of diazepam as a pre-anesthetic in combination with various anesthetics during orchidectomy in dogs. Agrobiological Records 3: 24-28. https://doi.org/10.47278/journal.abr/2020.021

Moerer M, Merle R, Bäumer W. A Cross-Sectional Study of Veterinarians in Germany on the Impact of the TÄHAV Amendment 2018 on Antimicrobial Use and Development of Antimicrobial Resistance in Dogs and Cats. Antibiotics (Basel). 2022 Apr 5;11(4):484.

4. Add future way forward in concluding remarks on the basis of outcomes antibiotics use in companion animal of this research.

5. Authors are advised to address above comments. 

Thanks 

Author Response

Please refer also to the new version of the manuscript as word-file (in track-change mode) and as pdf (without track-changes)

Reviewer 2 Report

Comments and Suggestions for Authors

The extensive use of antibiotics is a major factor in the development of resistant bacteria. Therefore, using antibiotics in animals may contribute to the potential risk of infections with resistant bacteria in humans, which are difficult to treat. Although many countries have systematically monitored and reduced antibiotic use in farm animals, there are no continuous monitoring analysis reports on antibiotic use in companion animals such as dogs and cats.

The manuscript is very well written. First, the references are comprehensive and well-summarized. The use of antibiotics in dogs and cats is also introduced at veterinary practices or Veterinary Teaching Hospital in some countries, such as the UK, Belgium, Italy, the Netherlands, and the US. Secondly, Antibiotic usage data in companion animals from 133 German practices from 2018 to 2022 were analyzed in detail from multiple perspectives. The title clearly reflects the main objective of the manuscript, and the abstract accurately summarizes the manuscript. The experimental design and methods are adequately described and appropriate for the study in the manuscript. The results and discussions are detailed and specific. In summary, this study can provide a scientific basis for establishing a monitoring system and serve as a robust tool for informing about legislative changes.

1. Line 107 missing F in the parentheses. 

2. Figure 3 is missing. However, there is a supplemental file attached; please clarify whether it is Figure 3 or another supplemental figure.

3. There is no figure legend below the supplemental figure.

4. Line 170 is miswritten; it should be 2.4 Influence factors for the use of antibiotics in dogs instead of 2.2.

Author Response

Dear reviewer, thank you for taking your time to review this manuscript. You can find our answers to your comments in the attachment.

Please refer also to the new version of the manuscript in word (in track-change mode) and in pdf (without track-changes)

Reviewer 3 Report

Comments and Suggestions for Authors

The manuscript and its syntax looks fine. However, there are few minor things that must be addressed before the manuscript can be considered for publication. Few of them are enlisted below:

1) Page 4, Line 107-Figure 3 is missing in the main text.

2) Ther manuscript needs to be checked for grammatical mistakes.

2.1) Page 2, Line 58 (no continuous monitoring of usage). Line 78-79 (25 practices never treated dogs).

2.2) In Section 5 [Conclusion], line 378-382 needs to be paraphrased. 

Author Response

Dear reviewer,

thank you for taking your time to review our manuscript and for your feedback. You can find the answers to your comments in the following. Please refer also to the new version of the manuscript in word (in track-change mode) and in pdf (without track-changes)

1) Page 4, Line 107-Figure 3 is missing in the main text.

Please excuse this mistake, we re-inserted the figure in the main text.

2) Ther manuscript needs to be checked for grammatical mistakes.

Thank you for this comments, we carefully revised the manuscript for grammatical mistakes.

2.1) Page 2, Line 58 (no continuous monitoring of usage). Line 78-79 (25 practices never treated dogs).

Thank you, we changed the sentences accordingly.

2.2) In Section 5 [Conclusion], line 378-382 needs to be paraphrased. 

Thank you for this comment. We rephrased as follows in ll. 387-391:

"Although there are some disadvantages to using routine data, as we lack some specific information, the amount of data that is available at a low cost can outweigh the disadvantages. And since the veterinarians document the treatments in order to create an invoice, the data is also complete and unbiased. With active monitoring, veterinarians might be more tempted to manipulate the data."